# Advances in Metabolomics: A Comprehensive Review of Type 2 Diabetes and Cardiovascular Disease Interactions

**DOI:** 10.3390/ijms26083572

**Published:** 2025-04-10

**Authors:** Lilian Fernandes Silva, Markku Laakso

**Affiliations:** 1Institute of Clinical Medicine, Internal Medicine, University of Eastern Finland, 70210 Kuopio, Finland; lilianf@uef.fi; 2Division of Cardiology, Department of Medicine, David Geffen School of Medicine, University of California, Los Angeles, CA 90095, USA; 3Department of Medicine, Kuopio University Hospital, 70200 Kuopio, Finland

**Keywords:** type 2 diabetes, cardiovascular disease, coronary artery disease, metabolomics, metabolites

## Abstract

Type 2 diabetes (T2D) and cardiovascular diseases (CVDs) are major public health challenges worldwide. Metabolomics, the exhaustive assessment of metabolites in biological systems, offers important insights regarding the metabolic disturbances related to these disorders. Recent advances toward the integration of metabolomics into clinical practice to facilitate the discovery of novel biomarkers that can improve the diagnosis, prognosis, and treatment of T2D and CVDs are discussed in this review. Metabolomics offers the potential to characterize the key metabolic alterations associated with disease pathophysiology and treatment. T2D is a heterogeneous disease that develops through diverse pathophysiological processes and molecular mechanisms; therefore, the disease-causing pathways of T2D are not completely understood. Recent studies have identified several robust clusters of T2D variants representing biologically meaningful, distinct pathways, such as the beta cell and proinsulin cluster related to pancreatic insulin secretion, obesity, lipodystrophy, the liver/lipid cluster, glycemia, and blood pressure, and metabolic syndrome clusters representing different pathways causing insulin resistance. Regarding CVDs, recent studies have allowed the metabolomic profile to delineate pathways that contribute to atherosclerosis and heart failure, as well as to the development of targeted therapy. This review also covers the role of metabolomics in integrated metabolic genomics and other omics platforms to better understand disease mechanisms, along with the transition toward precision medicine. This review further investigates the use of metabolomics in multi-metabolite modeling to enhance risk prediction models for predicting the first occurrence of major adverse cardiovascular events among individuals with T2D, highlighting the value of such approaches in optimizing the preventive and therapeutic models used in clinical practice.

## 1. Introduction

Type 2 diabetes (T2D) and its comorbidities have increased rapidly throughout the world. The International Diabetes Federation reported that 10.5% of the adult population (20–79 years) had diabetes in 2021 (https://www.idf.org (accessed on 1 March 2025)). By 2045, approximately 783 million people will have diabetes, and over 90% of people with diabetes will have T2D. Therefore, the identification of individuals with a high risk of T2D is of great importance.

Metabolomics, a comprehensive study of biological metabolites, is becoming increasingly important for understanding the mechanisms of metabolic diseases, including T2D and comorbidities, and especially cardiovascular diseases (CVDs) [1,2]. Biomarker discovery by metabolomics facilitates identification of the underlying metabolic perturbations associated with these complex multifactorial diseases and can provide valuable insights into their pathogenesis, progression, and therapeutic target [3,4].

The integration of metabolomics into clinical practice has been increasingly acknowledged as critical in clinical practice to improve diagnosis, prognosis, and therapy in patients manifesting such diseases [5,6]. The evolution of metabolomics in the context of T2D and CVDs highlights its role in biomarker discovery and risk prediction. The integration of metabolomic profiles into risk stratification models enhances the prediction of major adverse CVD events in patients with T2D. This shift toward personalized medicine underscores the need for understanding individual metabolic responses, allowing for further tailoring of prevention strategies and therapeutic approaches to these patients [7].

Although T2D and CVDs share several metabolic disturbances, the causal mechanisms involved in each of them are substantially different. The main characteristics of T2D are insulin resistance and impaired insulin secretion, leading to hyperglycemia, whereas CVDs are linked with chronic inflammation and oxidative stress, leading to atherosclerosis [8,9]. These differences require specific treatments that address the metabolic pathways involved in each of these diseases. The role of diet in ameliorating metabolic profiles has emerged as a promising area of therapeutic interest, with specific diets demonstrating potential benefits for both T2D and CVD risk factors [10,11,12].

Metabolomics studies alone are unable to obtain all the information needed. Therefore, metabolomics is often combined with genomics, proteomics, transcriptomics, and microbiome analysis. This comprehensive approach aims to understand the interactions between complex biological pathways and holds the potential for novel diagnostic techniques and therapies [13,14]. Among the omics techniques, metabolomics is considered the field most directly related to phenotypes, as metabolites act as direct regulators of biological processes [15]. As the global burden of T2D and CVDs continues to rise, particularly in developing countries, the ongoing exploration of metabolomics holds significant potential for improving health outcomes and informing precision medicine initiatives [3,7,16]. To support this review, a comprehensive literature search was conducted using the PubMed, Web of Science, and Scopus databases. Keywords such as “metabolomics”, “type 2 diabetes”, and “cardiovascular diseases” were used to identify relevant articles published up to March 2025, focusing on metabolic profiling, biomarker discovery, and precision medicine approaches.

## 2. Metabolomics

The metabolic state of an organism depends on the genome, transcriptome, proteome, epigenome, microbiome, and exposome (environment). Metabolomics, the analysis of small molecules (<1500 Da) within organisms, has a high potential to accurately describe the organism’s physiological state [17]. It allows the identification of diagnostic biomarkers, therapeutic targets, and personalized medical strategies, and is of great importance in the development of clinical research. Nuclear magnetic resonance (NMR) spectroscopy and mass spectrometry (MS) are the two major techniques applied in metabolomics, each with their strengths and weaknesses (Figure 1).

### 2.1. NMR Spectroscopy

NMR spectroscopy detects the magnetic properties of atomic nuclei to identify and quantify metabolites. It is a highly reproducible and non-destructive method with simple sample preparation. As a case in point, a 600 MHz NMR system can identify ∼400 metabolites per spectrum, offering a comprehensive window into the global metabolic fingerprint. Platforms like AXINON^®^ simplify system application [17], requiring minimal operator training compared to the complexity of MS workflows [18].

The reproducibility of NMR, with respect to laboratories and platforms, is one of its principal advantages. In addition, its non-destructive nature enables the preservation of samples for later use, a significant benefit compared to MS, where samples are destroyed in the process of ionization [18]. Along with its advantages, NMR also has disadvantages, such as lower sensitivity compared to MS, making it less suitable for detecting low-abundance metabolites [19,20]. It also requires large sample volumes and incurs high instrument costs, preventing wide-scale accessibility [21].

### 2.2. Mass Spectrometry

The ionization of metabolites is performed in MS to determine their mass-to-charge ratio. MS is widely known for its high sensitivity and specificity, which allows the detection of metabolites at trace levels [19,22]. MS also provides versatility, offering different ionization and separation techniques, and is a suitable tool for analyzing complex biological samples [23]. It is particularly beneficial for large-scale analyses due to its high-throughput abilities [21].

Along with its strengths, MS also has its challenges. Sample preparation can require many complex and time-consuming steps involving the use of specialized workflows that could elevate costs [21]. MS data are sometimes very complex to interpret because of their fragmentation patterns, requiring high-level training [19,20]. Additionally, matrix effects in biological samples can interfere with metabolite quantification, limiting the use of MS in less specialized environments [20].

### 2.3. Selection of the Method

Both approaches are widely used in clinical and biomarker research. For example, untargeted metabolomics approaches have employed MS to identify differential metabolites in patients with T2D [24]. In lipidomics, MS has also shown its power, a typical example being the establishment of high-coverage lipidomics protocols based on ultra-high-performance liquid chromatography coupled with MS [25]. NMR, while less sensitive, has been used for harmonizing metabolomics data through inter-laboratory comparisons, highlighting its strength during standardization [26]. In addition, NMR can analyze samples without preparative steps and in their native state, making it an important tool for assessing metabolic profiles [26].

The selection of NMR or MS is often tied to the research goals and the properties of the sample. NMR has the advantage of reproducibility and the additional feature of non-destructive analysis, whereas MS provides superior sensitivity and the availability of assays for a much wider range of metabolites [27]. In summary, these approaches are synergistic and make metabolomics a powerful tool to assess the human phenome. Empowered by these strengths, researchers may gain new insights into disease mechanisms and develop targeted, precision medical interventions [18,19,20,21,22,26].

## 3. Applications and Methodological Approaches of Metabolomics in Type 2 Diabetes

### 3.1. Biomarker Discovery and Risk Prediction in T2D

Recent studies demonstrate that metabolomics significantly enhances existing models for predicting major adverse cardiovascular events in patients with T2D. The SCORE2-Diabetes model predicts the 10-year risk of CVD when augmented with seven specific metabolites and substantially improves cardiovascular risk stratification, which is critical for developing personalized prevention measures in this high-risk population [5]. This shift toward personalized medicine highlights the importance of integrating metabolomics into clinical assessments.

In support of these efforts, recent studies employing high-throughput metabolomics have revealed metabolic signatures that may be predictive of T2D and its complications, including CVD. A wide range of plasma and serum metabolites, such as branched-chain and aromatic amino acids, acylcarnitines, ceramides, and carbohydrates, have been associated with an increased risk of T2D, while other metabolites, including glycine, glutamine, and indolepropionate, were associated with a reduced risk of T2D [28,29,30,31,32,33]. These signatures may emerge years before the clinical onset of disease, positioning metabolomics as a promising tool for early screening and risk stratification. In addition, metabolite profiles have been shown to differ, not only across stages of T2D development but also among diabetes subtypes and complications, as well as in response to pharmacological treatments. The integration of metabolomics with genomics and proteomics in multi-omics frameworks has further elucidated disease mechanisms and reinforced the potential for personalized therapeutic approaches. These findings collectively highlight the growing utility of metabolomics in advancing precision medicine for T2D and associated cardiovascular outcomes [28,29,30,31,32,33].

A study by Xie et al. [34] evaluated the added predictive value of metabolomic biomarkers for assessing the 10-year risk of T2D when combined with the clinical Cambridge Diabetes Risk Score. Using data from 86,232 UK Biobank participants and external validation in 4383 individuals from the German ESTHER cohort, 11 NMR-derived metabolites, including glycolysis-related metabolites, ketone bodies, amino acids, and lipids were identified. Adding these metabolites to the Cambridge Diabetes Risk Score significantly improved prediction accuracy, increasing the C-index from 0.815 to 0.834 in the UK Biobank cohort, and from 0.770 to 0.798 in the ESTHER cohort. The prediction accuracy of the continuous net reclassification index improved by 39.8% and 33.8%, respectively. A concise model with only four metabolites showed similar predictive performance. The UK Biobank Diabetes Risk Score demonstrated significant improvement over the clinical Cambridge Diabetes Risk Score and is well-suited for routine clinical use. Its reliance on minimal clinical information and low-cost NMR metabolomics highlights its potential for widespread application in diabetes risk assessment.

Despite these advances, developing robust prediction models using metabolomics data requires appropriate multivariate statistical methods. Techniques such as principal component analysis, cluster analysis, and machine learning are essential for handling high-dimensional data and extracting clinically relevant patterns. Incorporating these tools is crucial to improve the predictive utility of metabolomics with T2D-related cardiovascular outcomes [35].

### 3.2. Amino Acids and Metabolite Profiles in T2D

A well-known characteristic of T2D that was revealed by metabolomics is disturbed amino acid pathways. In patients with T2D, higher levels of branched-chain amino acids such as valine, leucine, and isoleucine, and of aromatic amino acids, including phenylalanine and tyrosine, are consistently observed [36]. However, there is no evidence that branched-chain amino acids are causally associated with insulin resistance and T2D [37]. In addition, proline, glutamate, and lysine levels were found to be elevated and glycine decreased in patients with T2D compared to healthy controls [7].

Other metabolomics studies identified specific signatures associated with T2D progression, including a metabolomic signature of increased α-hydroxybutyrate and inflammatory markers like interleukin that correlated with insulin resistance, independently of body mass index or age [7]. Chronic low-grade inflammation contributes to T2D pathogenesis by reprogramming macrophage metabolism, increasing glycolysis, and producing reactive oxygen species (ROS), which exacerbate insulin resistance [15,38]. Oxidative stress, which is caused by disrupted oxidative phosphorylation and the augmented generation of ROS, contributes to mitochondrial dysfunction, further damaging beta-cells in T2D. Moreover, hyperglycemia and hyperlipidemia are linked to epigenetic changes, including DNA hypermethylation, promoting a pro-inflammatory state [15]. For example, elevated levels of 25-hydroxycholesterol activate DNA methyltransferase-1, which causes epigenetic alterations and the dysregulation of carbohydrate and lipid metabolism, increasing the risk of T2D [6].

### 3.3. Applications of Metabolomics Risk Assessment in T2D

The identification of biomarkers, both genetic and non-genetic, through population-based studies has advanced our understanding of T2D heterogeneity, enabling improved disease classification and tailored interventions [38]. Biomarkers including mannose, alanine transaminase, uric acid, and genetic risk scores have been investigated for the identification of high-risk individuals and subgroups among patients with T2D, enhancing the application of precision medicine approaches. However, the limitations of genetic risk scores due to non-genetic influences highlight the need for comprehensive metabolic profiling and refined treatment strategies [39].

A systematic review and meta-analysis highlighted the association of 123 metabolites with the risk of T2D, based on data from over 71,000 participants [40]. Associations with increased risk were found for branched-chain amino acids (BCAAs), aromatic amino acids, carbohydrates, and lipids, including acylcarnitines and glycerolipids, while glycine, glutamine, and lysophosphatidylcholines were associated with reduced risk [40]. The dysregulation of the metabolic pathways highlighted in these findings suggests that metabolomics may be a new tool to identify clinically relevant biomarkers for T2D risk assessment. However, the observational nature of the studies and their inherent heterogeneity highlight the need for further validation [40].

A study on young adults in Finnish cohorts identified 113 metabolic measures predictive of T2D, with BCAAs, linoleic n-6 fatty acids, and lipoprotein measures emerging as strong indicators [40]. These observations provided evidence that the metabolic changes associated with T2D could be identified several years in advance of the clinical outcome and also illustrated the potential use of metabolomics for early treatment. A multi-metabolite score was created to demonstrate the power of aggregating metabolic biomarkers for accurate risk stratification [40].

An untargeted metabolomics study explored serum metabolomic fingerprints in obesity, insulin resistance, and T2D, identifying numerous metabolites, including the amino acids, lipids, fatty acids, and glycerol [41] associated with the development of T2D. These findings highlighted metabolic dysregulation in glucose metabolism, amino acid pathways, and lipid metabolism, offering insights into potential diagnostic and therapeutic targets for T2D. This study reinforced the role of metabolomics in advancing precision medicine for metabolic disorders [42].

A Korean cohort study including 1939 participants reported serum metabolites associated with the incidence of T2D in an 8-year follow-up. Alanine, arginine, isoleucine, proline, tyrosine, valine, hexose, and five phosphatidylcholine diacyls were identified as increasing the risk of T2D. Lyso-phosphatidylcholine acyl C17:0 and C18:2, along with other glycerophospholipids, decreased the risk of T2D, as well as a healthy diet. This study demonstrated the association between metabolic profiles and lifestyle factors, emphasizing the importance of dietary quality on the risk of T2D [43].

In summary, by integrating genetic and metabolic biomarkers and lifestyle factors, these studies collectively provide a deeper understanding of the mechanisms leading to an increasing risk of T2D, contributing to improved disease classification, risk prediction, and personalized treatment strategies. Biomarker identification through metabolomics offers a foundation for early diagnosis, prevention, and targeted interventions, paving the way for advancements in T2D management [7,38,39,40,41,42,43].

### 3.4. Mendelian Randomization Studies in T2D

Studies applying metabolomics have reported multiple associations of metabolites with several diseases, including T2D and CVD. The most reliable studies are those with a large number of participants being included and metabolites being measured. However, association studies do not prove causality. Mendelian randomization (MR) is a method that can strengthen causality by using genetic variants as instrumental variables [44]. Instrumental variables need to be associated with exposures, do not share a common cause with the outcome, and are only related to the outcome through exposure [5].

Yuan and Larsson applied a Mendelian randomization approach [45] to investigate risk factors for T2D. They found evidence for the causal associations of 19 risk factors previously published by the Diabetes Genetics and Meta-analysis Consortium (74,124 cases and 824,006 controls). Causal associations were found with insomnia, depression, systolic blood pressure, smoking initiation, lifetime smoking, coffee consumption, the metabolites isoleucine, valine, and leucine, liver alanine aminotransferase, childhood and adulthood obesity, visceral fat mass, resting heart rate, and four fatty acids.

A recent study [46] summarized those MR studies reporting causal associations revealing an increasing risk of T2D. These associations included increased systolic blood pressure [47], increased concentrations of liver function as measured by aspartate transferase and alanine transaminase [48], increased liver volume [49], increased obesity [50], increased visceral fat mass [51], and smoking [52]. Additionally, MR studies have also reported that increased concentrations of IGF-1 and circular protein biomarkers have causal associations with T2D [53].

### 3.5. Microbiome-Related Metabolites and the Risk of T2D

The gut microbiota plays a critical role in maintaining human health [54]. Gut microbiota-derived metabolites include bile acids, lipopolysaccharides, trimethylamine-N-oxide, tryptophan and indole derivatives, and short-chain fatty acids [55]. Short-chain fatty acids contribute to the metabolic regulation and energy homeostasis of the host, not only by serving as energy substrates but also by entering the systemic circulation as signaling molecules, affecting the gut–brain axis and neuroendocrine-immune network [56]. Additionally, short-chain fatty acids play an important role in the development of insulin resistance and T2D [57].

The gut microbial composition is significantly different between those patients with T2D and healthy subjects [58]. Gut microbes affect the host glucose metabolism through microbial metabolites, which are involved in diverse metabolic pathways. Changes in the metabolites produced by gut microbiota have been implicated in several diseases, including T2D, as well as metabolic syndrome [59,60]. Additionally, a recent study showed that the genetic associations between gut microbiota and T2D were mediated by plasma metabolites [61].

Follow-up studies are important to identify the microbiome metabolites associated with T2D. Our METSIM study included 5181 men [62]. In total, 4851 of them eventually participated in a 7.4-year follow-up visit, and 522 of them had developed T2D [63]. We identified several novel gut microbial metabolites that were significantly associated with an increased risk of developing T2D, including creatine, 1-palmitoleoylglycerol (16:1), urate, 2-hydroxybutyrate/2-hydroxyisobutyrate, xanthine, xanthurenate, kynurenate, 3-(4-hydroxyphenyl) lactate, 1-oleoylglycerol (18:1), 1-myristoylglycerol (14:0), dimethylglycine, and 2-hydroxyhippurate. These metabolites were associated with decreased insulin secretion or insulin sensitivity or both, suggesting mechanisms for the conversion of healthy sugar metabolism to T2D.

In our study, nine AAs (phenylalanine, tryptophan, tyrosine, alanine, isoleucine, leucine, valine, aspartate, and glutamate) were significantly (*p* < 5.8 × 10^−5^) associated with decreases in insulin secretion and the elevation of fasting or 2-h glucose levels. Tyrosine, alanine, isoleucine, aspartate, and glutamate were also significantly associated with the incidence of T2D after an adjustment for the known risk factors for T2D. Our study is the first with a population-based large cohort to report that AAs are associated not only with insulin resistance but also with decreased insulin secretion. Our study shows that microbial metabolites are important biomarkers for the risk of developing T2D.

Our results agree with previous findings showing a significant association of the BCAAs isoleucine, valine, and leucine with insulin resistance [64]. However, BCAAs were also associated with reduced insulin secretion in our study, suggesting that elevated concentrations of BCAAs may, over time, result in a decrease in insulin secretion. Impaired BCAA catabolism has been suggested to result in the accumulation of potentially toxic intermediates that contribute to β-cell mitochondrial dysfunction and eventually to the apoptosis of β-cells, which may lead to a decrease in insulin secretion [65].

### 3.6. Heterogeneity of T2D

T2D is a heterogeneous disease that develops through diverse pathophysiological processes and molecular mechanisms; therefore, the disease-causing pathways of T2D are incompletely understood [66]. Udler and collaborators were the first to investigate the genetic variants from genome-wide association studies to identify causal mechanisms for T2D [67]. They identified five robust clusters of T2D variants representing biologically meaningful distinct pathways, with the beta cell and proinsulin cluster related to pancreatic insulin secretion and the obesity, lipodystrophy, and liver/lipid cluster representing the different pathways causing insulin resistance.

A recent study by Suzuki et al. [68] included 2,535,601 individuals and 428,452 cases of T2D, aiming to understand the heterogeneity of T2D. They identified 1289 independent association signals at the genome-wide significance level that map to 611 loci [68]. They identified eight non-overlapping clusters of T2D signals that are characterized by distinct profiles of cardiometabolic trait associations. Among the clusters, five overlapped with the clusters identified by a previous study [68] and three were new, namely, the glycemia cluster (increased fasting glucose or hamoglobin), blood pressure cluster, and metabolic syndrome cluster. Clustering provides a framework to better understand the diverse physiological processes through which T2D develops.

Suzuki et al. [68] did not test the significance of the metabolites as biomarkers to understand those pathways crucial to the process of conversion to T2D. However, lipids and lipoproteins were important for building a cardiometabolic profile for metabolic syndrome: the obesity and lipodystrophy clusters (increased triglycerides and decreased high-density lipoprotein cholesterol) and the liver and lipid metabolism cluster (decreased triglycerides and high-density lipoprotein cholesterol). Elevated blood pressure was identified in metabolic syndrome and lipodystrophy clusters. Studying the etiological heterogeneity that drives the development and progression of T2D improves our understanding of the pathophysiological processes that link T2D to vascular outcomes.

### 3.7. Integrative Profiling and Future Directions in T2D

Figure 2 shows how the integrated profiling of metabolic biomarkers helps provide a broad and more complete view of the intricate signaling pathways involved in T2D, moving beyond isolated biomarker analysis. This methodology not only improves the identification of novel biomarkers but also enhances our understanding of the metabolic dysregulation associated with pre-diabetes and T2D. As the incidence of T2D continues to rise globally, particularly in developing regions like China and India, there is a pressing need for further research to establish metabolite profiles that can facilitate its early diagnosis and intervention strategies [7].

## 4. Metabolomics of Cardiovascular Diseases

CVDs, including coronary artery disease (CAD), myocardial infarction, heart failure, stroke, and hypertension, are the leading cause of death worldwide and impose a substantial socio-economic burden on healthcare. CAD is the most common manifestation of atherosclerosis; therefore, we focus mainly on CAD in this review. High concentrations of low-density cholesterol (LDL), elevated blood pressure, smoking, and T2D are the major risk factors for CAD. Our previous study suggested that patients with T2D without previous myocardial infarction are at as high a risk for myocardial infarction as nondiabetic individuals with previous myocardial infarction [69]. Therefore, T2D and the metabolic changes associated with this disease are major risk factors for CAD.

The initiation of atherosclerosis involves three processes: atherogenic lipid deposition, pro-inflammatory conditions, and endothelial dysfunction [70]. Improving risk stratification in clinical practice helps to combat this burden. Metabolites associated with CAD, heart failure, myocardial infarction, and stroke have emerged as crucial elements in understanding the pathophysiology of CVD. Metabolites have been linked to progression, outcomes, and risk stratification, and they can serve as powerful biomarkers, providing insights into disease mechanisms and potential therapeutic targets. Compounds such as trimethylamine N-oxide (TMAO) and phenylacetylglutamine [71] have received attention for their roles in exacerbating cardiovascular risk factors, highlighting the significance of metabolic disturbances in CAD. TMAO has been shown to promote platelet hyperreactivity and vascular inflammation, while phenylacetylglutamine impacts cardiac remodeling and left ventricular function. Both metabolites have been associated with increased risks of major adverse cardiovascular events and mortality, underscoring their relevance in clinical settings [72,73]. Plasma metabolomics reveals the shared and distinct metabolic disturbances associated with cardiovascular events.

Lv J. et al. [74] conducted an untargeted metabolomics for 333 participants with incident cardiovascular events and 333 matched controls. The CVD events were cardiovascular death, myocardial infarction, stroke, and heart failure. Metabolic pathway analysis unveiled 19 dysregulated metabolic pathways related to the composite of cardiovascular events, including tyrosine metabolism, cysteine and methionine metabolism, pentose and glucuronate interconversions, lysine degradation, and fatty acid biosynthesis. A total of 23 out of 333 participants shared different metabolites, mainly acylcarnitines, which were associated with the variable CVD events, suggesting heterogenous mechanisms across the different events.

The identification of metabolic signatures through advanced techniques, including metabolomics, has opened new avenues for predicting clinical outcomes, with studies demonstrating that metabolite-based models can outperform traditional clinical predictors in assessing the risks of mortality and morbidity [75]. The ongoing exploration of metabolites in the context of CVD highlights their role in understanding the metabolic remodeling that occurs during ischemic events. Metabolomics studies have linked specific compounds to adverse left ventricular remodeling and poor outcomes following a heart attack, paving the way for personalized approaches when managing cardiovascular diseases [72].

Despite advancements in the field, challenges remain, including the need for larger cohort studies and more sophisticated analytical methods to validate these findings and enhance their applicability in clinical practice [76]. One of the new approaches is from a recent study where the investigators developed a CAD-predictive machine learning model to build a risk score, based on the metabolite data from 93,642 individuals in the UK Biobank [77].

### 4.1. Metabolites Associated with CAD

Large population-based studies are needed to reveal potential biomarkers for CAD [78]. Lipids and their metabolites, particularly those involved in glycerophospholipid metabolism, have emerged as significant contributors to CAD [79]. Studies have shown that multiple lipid species are associated with inflammatory markers, including various phosphatidylcholines and lysophosphatidylcholines. The balance between saturated and unsaturated fatty acids in lipid composition influences their functional properties, with polyunsaturated species generally providing protective effects against inflammation and atherosclerosis [80].

Two modified nucleosides have been identified as significant metabolites in the context of CAD, namely, 2-dimethylguanosine and pseudouridine [80]. Both metabolites have correlated positively with the neutrophil-lymphocyte ratio and systemic immune inflammation index, indicating their potential role in atherosclerosis progression. This study is the first to report the contribution of pseudouridine to coronary atherosclerosis.

We found in the 12-year follow-up study of the METSIM cohort that nine amino acids, namely, alanine, glutamine, glycine, histidine, isoleucine, leucine, phenylalanine, tyrosine, and valine, were significantly associated with cardiovascular diseases [81]. Amino acids have also been reported to be associated with CVDs in other studies [82,83]. The amino acid derivative N6-acetyl-L-lysine has also been associated with the systemic immune inflammation index and the neutrophil-to-lymphocyte ratio, contributing to increased carotid atherosclerosis. This compound plays a crucial role in post-translational modification, influencing the function of key immune regulators like the NF-κB family. The enzymes responsible for regulating lysine acetylation, lysine acetyltransferase and deacetylases, present potential therapeutic targets in cardiovascular diseases [80].

Comprehensive metabolomic profiling has revealed several metabolites uniquely associated with inflammatory states in CAD including specific lipid species and derivatives such as ceramides, triglycerides, and various forms of phosphatidylcholines. These metabolites demonstrate varying relationships with inflammatory markers, highlighting their potential as biomarkers for disease progression and targets for therapeutic intervention [81]. By integrating the findings on these metabolites, it becomes evident that a complex interplay of nucleosides, amino acid derivatives, and lipid metabolites contributes to the pathophysiology of CAD, underscoring the importance of metabolomic and lipidomic studies for understanding cardiovascular health.

Omori et al. [84] identified seven metabolites associated with an increased risk of CAD incidence in diabetic patients. These metabolites were pelargonic acid, glucosamine, galactosamine, thymine, 3-hydroxybutyric acid, creatine, 2-aminoisobutyric acid, and hypoxanthine, which were all significantly decreased in patients with CAD. Their study included serum samples from 55 patients, 6 of whom had developed CAD by the time of a follow-up study. Larger cohorts and experimental validation are needed to confirm the results of the research.

Vernon et al. [85] investigated the association of the metabolites with CAD plaque phenotypes in 1002 patients from the BioHEART-CT study. Four metabolites showed significant links to CAD. Dimethylguanidino valeric acid was associated significantly with calcified plaque and obstructive CAD. Glutamate was associated with non-calcified plaque and phenylalanine, calcified plaque, and obstructive CAD. In contrast, TMAO was negatively associated with non-calcified plaque. Additionally, the lipid and nucleotide metabolic pathways were independently linked to CAD. These findings highlight potential metabolic biomarkers and pathways.

Deng et al. [86] identified 24 metabolites that are significantly associated with the incidence of CAD. Elevated metabolite concentrations were found for mannonate, imidazole propionate, acisoga, maleate, cysteine sulfinic acid, gluconate, glucuronide of piperine metabolite, N2, N2-dimethylguanosine, 3-methyl catechol sulfate, and 1-palmitoyl-2-oleoyl-GPC. Decreased metabolite concentrations were found for cysteine-glutathione disulfide, asparagine, 1,5-anhydroglucitol, serotonin, adenosine, homoarginine, S-methylcysteine, 3-bromo-5-chloro-2,6-dihydroxybenzoic acid, and 10-undecenoate.

Sheng et al. [87] performed a Mendelian randomization study to investigate the genetic associations between circulating metabolites (*n* = 24,925) and CAD (*n* = 86,995), using publicly available genome-wide association data. The results revealed that several lipid-related metabolites were positively associated with CAD, indicating an increased risk of CAD. Free cholesterol in large LDL (low-density lipoprotein), total cholesterol in medium LDL, and total cholesterol in LDL showed strong associations with CAD. These findings highlight the critical role of lipid metabolism in CAD risk and suggest potential metabolic targets for prevention and intervention.

### 4.2. Mechanisms Linking Metabolites to CAD

Metabolites play a significant role in the pathogenesis of CAD through various mechanisms, including inflammation, vascular dysfunction, and adverse cardiac remodeling [88]. Alterations in the metabolic pathways have been implicated in the progression of CAD, influencing both clinical outcomes and disease prognosis [72,89].

The relationship between inflammation and metabolism in CAD is especially important. Immune cells undergo metabolic changes that can influence their function and contribute to disease development [80]. For example, the metabolites associated with the urea cycle and oxidative stress have been linked to low-grade inflammation, which plays a crucial role in the atherosclerotic process [80]. Specific metabolites, such as 2-dimethylguanosine and pseudouridine, may serve as markers for vascular endothelial stress during CAD progression, while metabolites like hydrocinnamic acid exhibit anti-inflammatory properties that could mitigate cardiovascular risk [80].

### 4.3. Metabolomic Profiling and Disease Mechanisms in CAD

Metabolomic profiling reveals distinct metabolic signatures related to CAD [80]. Metabolites such as kynurenines, phenylacetyl-L-glutamine, and modified nucleosides increase the risk of death and major adverse CVD events by impairing cardiac function and promoting adverse left ventricular remodeling [89]. This underscores the potential of using metabolite profiles to improve risk stratification and guide therapeutic interventions in patients with CAD [72].

Several specific metabolites have been identified as playing critical roles in the progression of CAD. TMAO, a metabolite derived from dietary phosphatidylcholine, enhances platelet hyperreactivity and increases the risk of thrombosis, thereby promoting vascular inflammation [72]. Additionally, certain metabolites related to glycerophospholipid metabolism and amino acid metabolism, such as L-ornithine and L-glutamate, have been shown to correlate with inflammatory markers and could contribute to atherosclerosis [80].

The mechanisms by which metabolites influence cardiac remodeling in CAD involve the activation of pro-inflammatory pathways and the modulation of oxidative stress. Metabolites such as ceramides and triglycerides exhibit differential effects on inflammatory indices and may contribute to atherosclerosis through immune activation and lipid accumulation [80]. The integration of metabolic signatures with clinical risk factors has the potential to enhance our understanding of CAD progression and identify novel therapeutic targets for secondary prevention [72,89].

Mendelian randomization analysis indicated that 11 metabolites, namely, kynurenine, N6-succinyl adenosine, phenyllactate, DL-P-hydroxyphenyllactic acid, 3,3′,5-Triiodo-l-thyronine, adipic acid, S-(5-Adenosy)-l-homocysteine, TMAO, 4-acetamidobutyric acid, d-sorbitol, and phenylacetyl-l-glutamine, demonstrated causal associations with the risk of death, suggesting that these metabolites influence disease progression and outcomes through their role in left ventricle dysfunction [72]. The investigation of these mechanistic pathways is important as it may reveal potential therapeutic targets for the prevention of CAD-related complications. It has also been shown that gut microbially produced indole-3-propionic acid inhibits atherosclerosis [90].

Elevated LDL cholesterol concentration and T2D are associated with CAD. Interestingly, individuals on cholesterol-lowering statin medication have an increased T2D risk, while individuals with hypercholesterolemia have a reduced T2D risk [91], suggesting that there is a relationship between lipids and glucose. The STARNECT study investigators constructed network models from the STARNET study, based on seven cardiometabolic tissues obtained from CAD patients during coronary artery bypass grafting surgery [92]. They integrated gene expression, genotype, metabolomic, and clinical data to identify a glucose- and lipid-determining regulatory network that showed inverse relationships between lipid and glucose traits. The authors obtained similar inverse relationships of glucose and lipid concentrations in mice models. These results are important because they prove that the metabolic and cardiovascular pathways interact in human metabolism.

## 5. Comparative Analysis Between Type 2 Diabetes and Cardiovascular Diseases

Metabolomics has emerged as a crucial field for understanding the metabolic dysregulation that is associated with complex diseases such as T2D and CVD. By examining the profile of these metabolites, it is possible to identify specific biomarkers that reflect disease states, aiding in diagnosis and potential treatment strategies.

Sex differences in metabolomics and proteomics are crucial for understanding the pathophysiology and clinical outcomes of T2D and atherosclerotic cardiovascular disease. Studies show that women with diabetes often face a higher cardiovascular risk than men, partly due to their distinct metabolic and proteomic profiles, which are influenced by hormonal factors such as estrogen and testosterone [30,93,94]. These differences affect biomarker expression, disease progression, and treatment response, underscoring the need to consider sex as a biological variable in risk assessment and therapy [95,96,97,98,99]. Identifying sex-specific biomarkers may enable more personalized and effective interventions to prevent cardiovascular complications in diabetic patients [100,101,102]. However, most clinical trials have disproportionately involved male participants, limiting the generalizability of these findings. Addressing this gap through inclusive, sex-sensitive research is vital for advancing precision medicine in T2D and atherosclerotic cardiovascular disease [103,104,105].

Lu et al. [106] examined the shared and unique associations of metabolites with T2D, CAD, and stroke. A total of 168 plasma metabolites were measured with high-throughput NMR among 98,162 participants without T2D, CAD, and stroke at baseline. Over 12.1 years of follow-up most lipoprotein metabolites were associated with the risk of both T2D and CAD but not with the risk of stroke. Phospholipids within intermediate-density lipoprotein or large low-density lipoprotein particles showed positive associations with CAD and negative associations with T2D. Metabolites indicating very small, very low-density lipoprotein, histidine, creatinine, albumin, and glycoprotein acetyls were associated with the risk of developing all three conditions. This large-scale metabolomics study revealed common and distinct metabolic biomarkers for T2D, CVD, and stroke.

Other studies have reported similar results as Lu et al. [106], suggesting that T2D and CVDs exhibit alterations in specific metabolites, reflecting underlying metabolic disturbances [107]. For instance, common metabolites such as amino acids and lipids have been implicated in the pathogenesis of both T2D and CVDs. Low concentrations of glycine and serine have been associated with T2D, whereas dysregulated lipid profiles are also observed in CVD patients [7,108]. These shared metabolic markers indicate the overlapping of metabolic pathways and suggest that interventions aimed at correcting these dysregulations may be of benefit for both conditions (Figure 3).

Despite their similarities, T2D and CVDs are driven by distinct metabolic mechanisms. T2D primarily involves insulin resistance and impaired glucose metabolism, leading to alterations in energy balance and fat storage [10]. In contrast, CVDs are often associated with chronic inflammation, oxidative stress, and lipid accumulation, resulting in atherosclerosis and other cardiovascular complications [10,16]. The unique metabolic alterations in each disease highlight the need for tailored therapeutic approaches. For example, while lifestyle interventions like diet modification may enhance metabolic profiles in patients with T2D, they may also play a critical role in reducing cardiovascular risk factors in patients with CVD [7,8,9,10,11,12,13,14,15].

## 6. Microbiota and Cardiovascular Diseases

The potential role of infectious microorganisms, including bacteria and viruses, as risk factors for CVD was discovered in epidemiological studies [108]. A microbiome is a collection of symbiotic microorganisms and their associated genomes. There are >38 trillion bacterial cells in each human body, which is far more than the total number of human cells. The majority of the microbiome, numbering approximately 10–100 trillion microbial cells, is located in the gastrointestinal tract [109]. The gut microbiome consists of 2172 species within 12 different phyla, most of which belong to the phyla *Proteobacteria*, *Firmicutes*, *Actinobacteria*, *Bacteroidetes*, and *Verrucomicrobia*. *Firmicutes* and *Bacteroidetes* are the two dominating microbes, accounting for more than 90% of the population [109,110]. Infection and inflammatory processes induce the onset, progression, and rupture of atherosclerotic plaques [111].

A study by Talmor-Barkan et al. [112] reported that patients with CAD had distinct metabolome and gut microbial signatures compared with the controls and were depleted in a previously unknown bacterial species of the *Clostridiaceae* family. Interestingly, this bacterial species was associated with concentrations of multiple circulating metabolites in controls, several of which have previously been linked to an increased risk of CAD. However, it is not entirely clear whether these bacterial species are causal for CAD.

The methods used to sequence the microbiome include amplicon sequencing, shotgun metagenomics, and RNA sequencing. Key differences between these techniques are the amount of genetic material that is sequenced and the resolution and coverage at which they can differentiate between microbial species. The majority of the first microbiome studies conducted analyses of the 16S rRNA gene [113]. However, 16S rRNA gene sequencing techniques are often not able to achieve the necessary resolution; therefore, such analyses are not able to identify less abundant microbial taxa.

## 7. Clinical Implications and Future Directions

The identification of metabolomics-based biomarkers in both T2D and CVDs holds promise for improving the early diagnosis and monitoring of disease progression. However, the current reliance on single biomarkers may limit the specificity and predictive power of diagnostic tests [15]. While metabolomics holds significant promise for improving early diagnosis, risk prediction, and the individualized treatment of T2D and CVD, its integration into routine clinical practice remains limited. Key challenges include methodological complexity, high implementation costs, and the lack of standardized protocols across laboratories and platforms. These limitations hinder the clinical translation of metabolomic findings and the development of universally applicable diagnostic tools. Future efforts should prioritize the creation of cost-effective, reproducible workflows and the validation of biomarker panels in diverse populations to support broader clinical adoption. Future research should focus on developing integrated profiles of multiple biomarkers to enhance diagnostic accuracy and develop more effective preventive strategies. Clinical trials incorporating these metabolomic insights are essential for validating the role of metabolites in managing T2D and CVDs, potentially leading to innovative therapeutic targets and interventions [2,7].

The future of metabolomics presents significant opportunities for advancing the understanding and management of metabolic diseases, particularly T2D and CVDs, through integrated multi-omics approaches (Figure 4). By combining metabolomics with genetics, transcriptomics, and proteomics, researchers can construct a more comprehensive picture of the metabolic pathways, leading to the improved identification of therapeutic targets and biomarkers for disease prevention and treatment [10,11].

A critical challenge in metabolomics is advancing data integration. Standardized protocols for data collection, processing, and analysis should be developed. This includes addressing sample variability, harmonizing data formats, and refining correlational analysis across omics datasets. Overcoming these barriers will improve the ability to link metabolites with disease states and clinical outcomes, ultimately paving the way for more personalized treatment strategies in T2D and CVDs [10,11].

Another essential aspect of future research is database development, which involves creating large-scale repositories that aggregate multi-omics data from diverse populations. Such databases will provide a foundation for population-based studies, helping researchers to understand how genetic, environmental, and lifestyle factors interact to influence metabolic health. Rigorous scientific screening and the integration of these datasets are essential for deriving meaningful insights [2,10].

Moreover, therapeutic interventions derived from metabolomics research must be translated into clinical applications. Validating the findings from multi-omics analyses through targeted molecular biology studies is crucial for identifying novel treatment strategies and preventive measures for T2D and CVDs [10,15]. The need to address health disparities in metabolic diseases is also gaining research attention. Understanding the variations in metabolic responses across different ethnic groups can enhance precision medicine strategies, ensuring that healthcare solutions are equitable and effective for all populations. By identifying unique metabolic profiles in diverse ethnic groups, researchers can develop tailored interventions that better serve specific community needs [2,114].

In summary, metabolomics continues to evolve, offering powerful insights into disease mechanisms, biomarker discovery, and precision medicine. Future advancements in data integration, database development, therapeutic applications, and health equity will be key to transforming metabolomics from a research tool into a clinical cornerstone for managing metabolic diseases. A summary of the key findings presented in this review regarding metabolomic profiling in T2D and CVD is provided in Table 1.

## Figures and Tables

**Figure 1 ijms-26-03572-f001:**
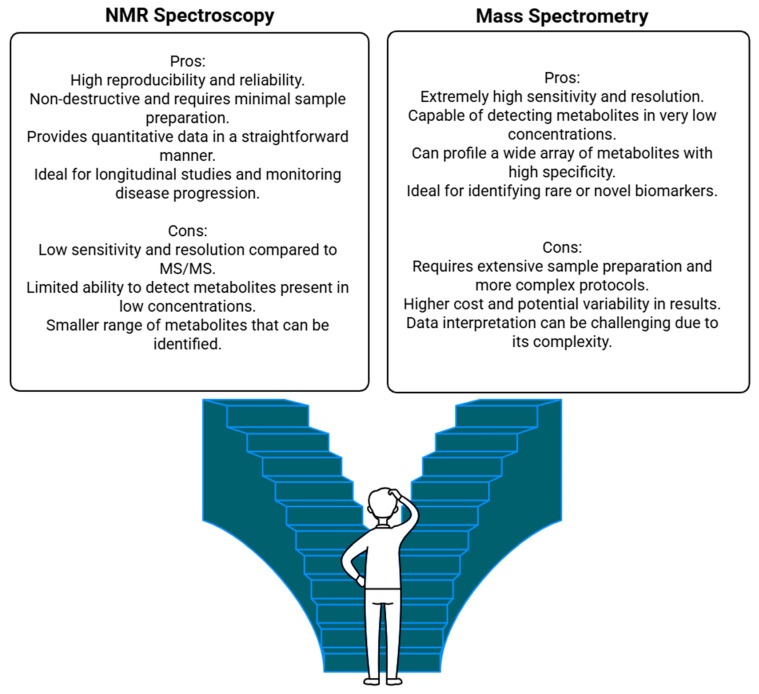
Main strengths and limitations of NMR spectroscopy and mass spectrometry techniques.

**Figure 2 ijms-26-03572-f002:**
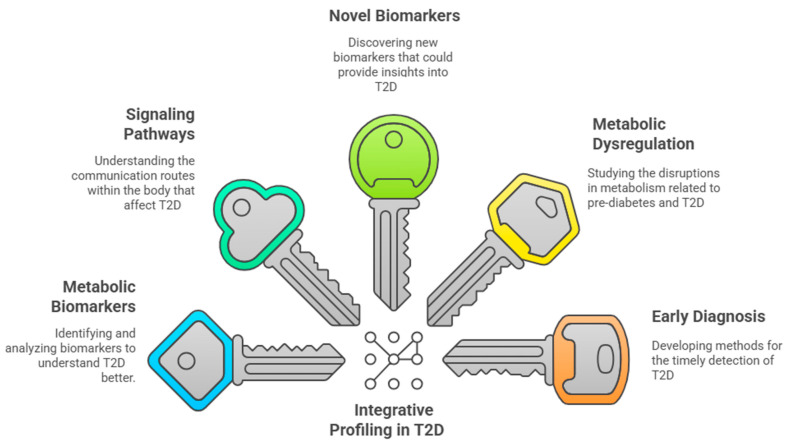
Integrative profiling and the future directions of integrative metabolic profiling in T2D.

**Figure 3 ijms-26-03572-f003:**
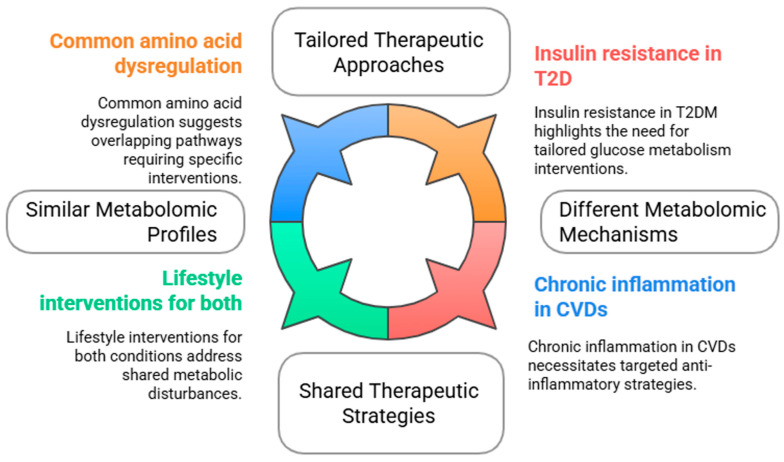
Comparative analysis of type 2 diabetes and cardiovascular diseases.

**Figure 4 ijms-26-03572-f004:**
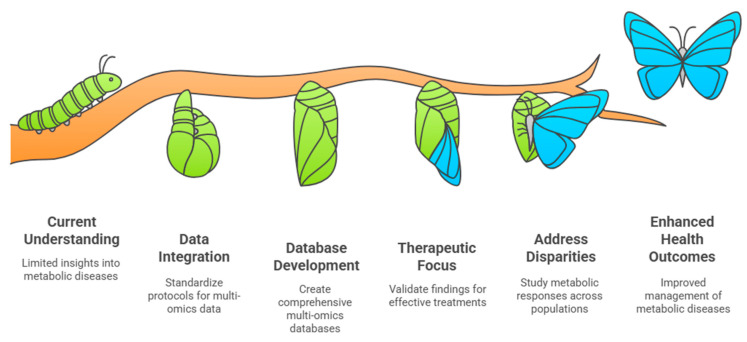
Future advancements in metabolomics, aiming at integration into clinical practice.

**Table 1 ijms-26-03572-t001:** Summary of the key findings in this review on metabolomics in T2D and CVDs.

	Key Findings	Implications
Biomarker Discovery and Risk Prediction	Specific metabolites (e.g., BCAAs, aromatic AAs, acylcarnitines, and ceramides) are associated with increased T2D risk, while glycine, glutamine, and indolepropionate are protective.	Enhances the early detection and risk stratification of T2D. Enables more accurate, personalized preventive strategies.
Amino Acid and Lipid Metabolism	Alterations in the amino acid and lipid pathways are consistently observed in T2D patients: BCAAs and aromatic AAs are elevated, while glycine is decreased.	Indicates metabolic dysregulation; there is the potential for therapeutic targeting and metabolic pathway modulation.
Metabolomics and CVD Risk	Metabolites such as TMAO, phenylacetylglutamine, and acylcarnitines are linked to CAD and heart failure risk.	Facilitates the identification of high-risk individuals and supports targeted interventions for cardiovascular outcomes in T2D patients.
Sex Differences	Metabolic and proteomic profiles differ by sex due to hormonal influences (e.g., estrogen and testosterone); women with T2D may face higher CVD risk.	Highlights the need for sex-specific risk assessment and therapeutic approaches.
Microbiota-Related Metabolites	Gut microbial metabolites (e.g., SCFAs and indole derivatives) are associated with insulin secretion, resistance, and T2D risk.	Emphasizes the gut–metabolism axis and its relevance in disease progression and treatment design.
Genetics and Precision Medicine	GWAS and polygenic risk scores identify clusters of T2D risk related to insulin secretion and resistance. Integration with metabolomics improves prediction.	Supports the implementation of precision medicine through integrated multi-omics profiling.
Clinical Challenges	Methodological complexity, lack of standardization, and cost limit metabolomics’ clinical adoption.	Necessitates development of standardized, cost-effective protocols and validation in diverse cohorts.

## Data Availability

No new data were created or analyzed in this study.

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
