# Peer review of "Advances in Metabolomics: A Comprehensive Review of Type 2 Diabetes and Cardiovascular Disease Interactions"

_ijms, 2025, doi:10.3390/ijms26083572_

Round 1
Reviewer 1 Report
Comments and Suggestions for Authors
The manuscript (ijms-3547192) is a further review article about the use of metabolomics in multi-metabolite modeling to enhance risk prediction models for the occurrence of major adverse cardiovascular events among individuals with T2D, highlighting the value of such approaches in optimizing preventive and therapeutic models in clinical practice. The authors are well engaged in the research areas and have published a number of key publications, including several review articles on the topic. The paper is well written and significantly contributes to the field of metabolomics by synthesizing current knowledge and highlighting the potential of metabolomics in clinical research and personalized medicine.
I have just a few suggestions for possible revision.
- Since this is a rapidly evolving and frequently reviewed research field (PubMed: Metabolomics and Diabetes and Cardiovascular Disease - 317 published reviews, Metabolomics and Proteomics in Type 2 Diabetes - 1026 published reviews), and a number of new review papers have not been cited and discussed in this review, it would be great to highlight in the abstract what recent advances can help in optimizing preventive and therapeutic models in clinical practice. Here are the papers not being discussed in this review paper: Curr Diab Rep. 2022 February; 22(2): 65–76; Diabetes Care 2022; 45: 1013–1024; Circ Res. 2020 May 22; 126(11): 1613–1627; Nutrients 2022, 14, 3201; Int. J. Mol. Sci. 2019, 20, 2467; J Diabetes Investig 2023; 14: 503–515.
- Metabolic and glycemic health sex dimorphism has been implicated in understanding differences in genetic effects between women and men in diabetes and atherosclerosis, as indicated by an increased number of published studies. It would be important to discuss sex differences in metabolomics and proteomics in diabetes and atherosclerotic cardiovascular disease and related therapeutic models in clinical practice.
- The manuscript might discuss potential biases due to an overemphasis on the advantages of metabolomics without addressing the difficulties faced in clinical implementation.
Author Response
I am attaching the answers for the reviewer.

Reviewer 2 Report
Comments and Suggestions for Authors
In the review “Advances in Metabolomics: A Comprehensive Review of Type 2 Diabetes and Cardiovascular Disease Interactions” the authors did not specify the title of chapter 3.
The authors did not specify which methodological approach they used to collect the data.
The authors should include a table summarizing the kay findings from the literature.
----
Additional comments:
The authors present the use of metabolomics in multi-metabolite modelling in individuals with cardiovascular diseases associated with type 2 diabetes.
This is a very interesting topic, especially considering that CVD is one of the most prevalent problems and unfortunately is often associated with T2D. The author gave an overview of the literature on these two diseases.
The integration of metabolomic profiles into risk stratification models could be very important to better understand the association between CVD in patients with T2D.
In order to make prediction, multivariate statistical methods must be used, which was not done in this paper, making it difficult to speak of prediction. Therefore, it is necessary to address this.
The authors did not specify in the methodology which database they used for data collection.
The authors did not specify the title of chapter 3.
The references are appropriate.
The author concludes that future advances in the integration of data, development databases, and therapeutic applications. The authors should include a table summarising the key findings from the literature.
The results presented in Figures 1-4 are clear and nicely.
Author Response
I am attaching the comments for the reviewer.
